# Association between *SMAD4* Mutations and *GATA6* Expression in Paired Pancreatic Ductal Adenocarcinoma Tumor Specimens: Data from Two Independent Molecularly-Characterized Cohorts

**DOI:** 10.3390/biomedicines11113058

**Published:** 2023-11-15

**Authors:** Joshua D. Greendyk, William E. Allen, H. Richard Alexander, Toni Beninato, Mariam F. Eskander, Miral S. Grandhi, Timothy J. Kennedy, Russell C. Langan, Jason C. Maggi, Subhajyoti De, Colin M. Court, Brett L. Ecker

**Affiliations:** 1Rutgers New Jersey Medical School, Rutgers Health, Newark, NJ 07103, USA; jdg221@njms.rutgers.edu (J.D.G.); wea13@njms.rutgers.edu (W.E.A.); 2Rutgers Cancer Institute of New Jersey, Rutgers Health, New Brunswick, NJ 08901, USA; ha364@cinj.rutgers.edu (H.R.A.); tmb171@cinj.rutgers.edu (T.B.); me550@cinj.rutgers.edu (M.F.E.); mg1354@cinj.rutgers.edu (M.S.G.); tk431@cinj.rutgers.edu (T.J.K.); russell.langan@rwjbh.org (R.C.L.); jason.maggi@rwjbh.org (J.C.M.); sd948@cinj.rutgers.edu (S.D.); 3Robert Wood Johnson Medical School, Rutgers University, New Brunswick, NJ 08901, USA; 4Cooperman Barnabas Medical Center, Livingston, NJ 07039, USA; 5Department of Surgical Oncology, University of Texas San Antonio, San Antonio, TX 78249, USA; courtc@uthscsa.edu

**Keywords:** pancreatic ductal adenocarcinoma, *SMAD4*, *GATA6*, basal-like, precision oncology, molecular biomarker

## Abstract

Several molecular biomarkers have been identified to guide induction treatment selection for localized pancreatic ductal adenocarcinoma (PDAC). *SMAD4* alterations and low *GATA6* expression/modified “Moffitt” basal-like phenotype have each been associated with inferior survival uniquely for patients receiving 5-FU-based therapies. *SMAD4* may directly regulate the expression of *GATA6* in PDAC, pointing to a common predictive biomarker. To evaluate the relationship between *SMAD4* mutations and *GATA6* expression in human PDAC tumors, patients with paired *SMAD4* mutation and *GATA6* mRNA expression data in the TCGA and CPTAC were identified. In 321 patients (TCGA: n = 180; CPTAC: n = 141), the rate of *SMAD4* alterations was 26.8%. The rate of *SMAD4* alteration did not vary per tertile of normalized *GATA6* expression (TCGA: *p* = 0.928; CPTAC: *p* = 0.828). In the TCGA, *SMAD4* alterations and the basal-like phenotype were each associated with worse survival (log rank *p* = 0.077 and *p* = 0.080, respectively), but their combined presence did not identify a subset with uniquely inferior survival (*p* = 0.943). In the CPTAC, the basal-like phenotype was associated with significantly worse survival (*p* < 0.001), but the prognostic value was not influenced by the combined presence of *SMAD4* alterations (*p* = 0.960). *SMAD4* alterations were not associated with poor clinico-pathological features such as poor tumor grade, advanced tumor stage, positive lymphovascular invasion (LVI), or positive perineural invasion (PNI), compared with *SMAD4*-wildtype. Given that *SMAD4* mutations were not associated with *GATA6* expression or Moffitt subtype in two independent molecularly characterized PDAC cohorts, distinct biomarker-defined clinical trials are necessary.

## 1. Introduction

Pancreatic ductal adenocarcinoma (PDAC) has one of the highest case-specific mortality rates of all cancers [1]. Although resection remains the only curative therapy for PDAC, improvements in long-term survival are attributable to advances in systemic treatment [2,3,4,5]. Currently, 5-fluorouracil-based (i.e., with irinotecan and oxaliplatin as FOLFIRINOX) or gemcitabine-based (with Nab-paclitaxel) chemotherapies are both used, with selection largely driven by patient-related factors such as age, comorbidity, and performance status. Amid the expanding options for systemic therapy and the mounting emphasis on administering such agents in the neoadjuvant setting, identification of biomarkers to guide first-line treatment selection remains a critical unmet need.

Previous work has identified genomic alterations in *SMAD4* as predictive of unique resistance to FOLFIRINOX induction. *SMAD4* alterations, primarily loss-of-function mutations [6], are present in approximately 20% of localized PDAC patients and may be linked to increased rates of metastatic progression and lower rates of surgical resection in patients receiving induction FOLFIRINOX but not receiving gemcitabine/nab-paclitaxel [7,8]. Separately, the modified Moffitt “basal-like” phenotype, marked by the loss of expression of *GATA6*, may also confer resistance to FOLFIRINOX chemotherapy [9,10,11,12,13]. In an exploratory analysis of the ESPAC-3 trial, low *GATA6* expression was associated with inferior clinical outcomes uniquely for patients receiving 5-FU/LV, but not gemcitabine [9]. Similarly, in the COMPASS trial of patients with locally advanced or metastatic PDAC receiving FOLFIRINOX or gemcitabine/nab-paclitaxel, those with a modified “basal-like” phenotype had uniquely fewer responses and worse overall survival when treated with FOLFIRINOX [13]. Moreover, *GATA6*-low cell lines derived from patient-derived xenografts were particularly resistant to 5-FU but not gemcitabine [9].

Understanding the relationship between *SMAD4* and *GATA6* will be important for informing future studies of molecular biomarkers to guide induction chemotherapy selection and rational clinical trial design. *SMAD4* is on chromosome 18q11, and 28.7 Mb from *GATA6*. *GATA6* and *SMAD4* are frequently co-altered [9]. Moreover, *SMAD4* may directly regulate the expression of *GATA6* [10]. Together, these data raise the possibility that *SMAD4* alterations at the genome level and loss of *GATA6* expression may represent a single molecular pathway that confers unique resistance to 5-FU-based therapies. In this study, representing the largest cohort of PDAC patients with paired DNA and RNA expression data, we examined the association between *SMAD4* alterations and *GATA6* expression to further characterize the translational potential of this molecular relationship. We hypothesized that *SMAD4* alterations would be positively correlated with *GATA6* expression and the basal-like subtype.

## 2. Methods

The study was deemed IRB exempt with use of publicly available and deidentified datasets. The cBioPortal platform for Cancer Genomics database is an open-access resource for exploration of multidimensional cancer genomics data. The platform was queried for PDAC samples with paired mutation and mRNA expression data, with identification of two datasets: the Cancer Genome Atlas (TCGA) and the Clinical Proteomic Tumor Analysis Consortium (CPTAC) [14].

*SMAD4* was considered altered when there was either a mutation or copy number deletion. The modified Moffitt phenotypes were assigned as previously described [15], where the R package ConsensusClusterPlus27 [16] was employed to subtype PDAC samples according to the expression signatures defined in Moffitt et al. [12]. Briefly, the number of clusters was confirmed by examining cumulative distribution function, with the existence of well-separated clusters for Moffitt et al. classification based on tumor (basal-like and classical) and stroma signatures. *GATA6* mRNA expression z-scores, as a surrogate biomarker of the modified Moffitt “basal-like” phenotype [13,17], were downloaded along with *SMAD4* alteration calls and paired clinical data from the FireBrowse data portal (http://firebrowse.org) accessed on 12 August 2022 (TCGA data version 2016_01_28) and from cbioportal (https://www.cbioportal.org) (CPTAC data) [14].

### Statistical Analysis

Descriptive statistics are presented as frequencies for categorical variables and median interquartile range (IQR) for continuous variables. Pearson’s χ^2^ and Wilcoxon rank-sum test were used to analyze categorical and continuous variables, respectively. *GATA6* mRNA expression z-scores were analyzed by tertiles in each study to evaluate low, medium, and high expression as previously described [18]. The primary outcomes assessed were (1) rates of *SMAD4* alterations for classical vs. basal-like subtypes and (2) rates of *SMAD4* alterations for *GATA6*-low vs. *GATA6*-medium/high. The secondary outcomes were the impact of *SMAD4* and molecular subtype on overall survival (OS), which was evaluated by Kaplan–Meier estimates. The variables associated with OS on univariable analysis (*p* < 0.10) were entered into a multivariable Cox regression model (TCGA: AJCC stage, nodal status, and tumor grade; CPTAC: AJCC stage). *p*-values ≤ 0.05 were considered statistically significant; all tests were two-sided. Analyses were carried out using SPSS v27.0 (IBM Corp., Armonk, NY, USA).

## 3. Results

### 3.1. Patient Cohort

In total, 321 patients with PDAC with paired DNA and RNA data were identified. The TCGA (n = 180) included 60 patients (33.3%) with *SMAD4* alterations (Table 1). The median age was 65 (IQR 56-73) and most patients were male (n = 99; 55.0%) and white (n = 158; 87.8%). Nearly all patients (n = 152; 94.4%) had AJCC pathologic stage I–II disease. Patients with and without *SMAD4* alterations were comparable with respect to age, sex, T-stage, N-stage, and tumor grade.

The CPTAC (n = 141) included 26 patients (18.4%) with *SMAD4* mutations (Table 2). The median age was 65 (IQR 60–71) and most patients were male (n = 74; 52.5%). Nearly all patients (n = 125; 93.3%) had localized (AJCC pathologic stage I–III) disease. Patients with and without *SMAD4* alterations were comparable with respect to age, T-stage, N-stage, and the presence of peri-neural or lymphovascular invasion.

### 3.2. Association between SMAD4 Alteration Status and Basal Subtype

In the TCGA cohort, the rate of *SMAD4* alteration did not vary per tertile of normalized *GATA6* expression (31.7% vs. 33.3% vs. 35.0%, *p* = 0.928) (Table 3). Likewise, the rate of *SMAD4* alteration was not associated with Moffitt subtype (classical: 37.2% vs. basal-like: 39.4%, *p* = 0.783). In the CPTAC cohort, the rate of *SMAD4* mutation did not vary per tertile of normalized *GATA6* expression (17.0% vs. 17.0% vs. 21.3%, *p* = 0.828). Similarly, the rate of *SMAD4* alteration was not associated with Moffitt subtype (classical: 22.5% vs. basal-like: 16.7%, *p* = 0.416).

### 3.3. Impact of SMAD4 and Moffitt Subtype on Long-Term Survival

In the TCGA cohort, patients were followed for a median of 24.2 (IQR 15.2–42.3) months. *SMAD4* alterations were not associated with significantly worse survival (estimated mean OS: 25.1 [95% CI 17.6–32.7] vs. 40.6 [95% CI 32.3–48.9] months, log rank *p* = 0.077; Figure 1A). Likewise, the basal-like phenotype was not associated with significantly worse survival (estimated mean OS: 20.4 [95% CI 16.3–24.5] vs. 30.8 [95% CI 24.4–37.2] months, log rank *p* = 0.080; Figure 1B). There were no trends in OS for *SMAD4* alterations in the subsets of the classical subtype (log rank *p* = 0.164; Figure 1C) or basal-like subtype (log rank *p* = 0.934; Figure 1D). *SMAD4* alterations were not associated with OS in a multivariable model accounting for AJCC stage, nodal status, and tumor grade (HR 1.25, 95% CI 0.82–1.91).

In the CPTAC cohort, patients were followed for a median of 23.9 (IQR 15.5–34.9) months. *SMAD4* alterations were not associated with worse survival (estimated mean OS: 23.3 [95% CI 18.1–28.5] vs. 21.2 [95% CI 18.0–24.5] months, log rank *p* = 0.277; Figure 2A). The basal-like phenotype was associated with significantly worse survival (estimated mean OS: 15.3 [95% CI 12.4–18.1] vs. 28.2 [95% CI 23.7–32.8] months, log rank *p* < 0.001; Figure 2B). There were no trends in OS for *SMAD4* alterations in the subsets of the classical subtype (log rank *p* = 0.359; Figure 2C) or basal-like subtype (log rank *p* = 0.960; Figure 2D). SMAD4 alterations were not associated with OS in a multivariable model accounting for AJCC stage (HR 0.75, 95% CI 0.39–1.42).

## 4. Discussion

Improving patient outcomes in pancreatic cancer hinges on precise chemotherapy selection to match tumor responsiveness. FOLFIRINOX has emerged as one of the most effective chemotherapeutic regimens for managing PDAC, demonstrating efficacy in both metastatic and adjuvant therapy settings [4,19]. As its application in the neoadjuvant setting continues to evolve, the integration of predictive biomarkers will be critical in refining treatment selection. *SMAD4* mutations and the basal-like expression subtype (or *GATAT6* low expression) have each emerged as potential biomarkers in this context. Herein, we present the largest cohort of PDAC patients evaluating the relationship between *SMAD4* mutation, the modified Moffitt phenotype, and *GATA6* expression in the context of clinical outcomes. Contrary to hypothesis, there was no correlation between *SMAD4* alterations and the modified Moffitt phenotype or *GATA6* expression. While in vitro data suggested that SMAD4 may regulate the expression of GATA6, these results are not confirmed in two human PDAC cohorts. Moreover, the combined presence of both biomarkers did not identify a patient subset with uniquely inferior outcomes. These data highlight the need for distinct biomarker-driven clinical trials and independent investigation of each biomarker in its potential mechanisms of treatment resistance.

*SMAD4* serves as a mediator in the TGFB1 (TGF-β) signaling pathway and is recognized as a driver of the progression of pancreatic intraepithelial neoplasia to invasive adenocarcinoma [20,21]. Together with *KRAS*, *TP53*, and *CDKN2A*, *SMAD4* is recognized as one of the four driver mutations in PDAC [17,20,22]. In these data, the rate of *SMAD4* alterations ranged from 18.4% to 33.3%. The rate of *SMAD4* in this study similarly compares to the previous literature’s rates of 20–33% [23,24]. This range likely represents inherent differences in study populations, where rates of *SMAD4* alterations increase with greater tumor burden. Iacobuzio-Donahue et al. previously observed that locally advanced PDAC without metastatic disease uncommonly showed loss of *SMAD4* (22%) as compared with carcinomas with extensive metastatic burden, where the rates of *SMAD4* alteration approached 75% [25].

*SMAD4* alteration has been reported to be an independent prognostic factor for recurrence-free survival and overall survival [26,27,28,29]. In a meta-analysis of eight studies with available data on *SMAD4* status and patient survival, *SMAD4* alterations conferred a pooled 40% increased risk of death [30]. Several studies have observed that the loss of *SMAD4* is linked to distant metastases, which may explain its prognostic significance [25,31]. In our data, there was a trend to inferior survival in patients with *SMAD4* alterations in the TCGA. However, a similar relationship was not observed in the CPTAC cohort. Notably, some studies have not identified *SMAD4* as a prognostic biomarker [32,33]. Winter et al. reported that loss of *SMAD4* expression showed no association with either recurrence or early mortality in resected PDAC patients [23]. These conflicting data highlight that disease-related survival is a complex interplay not solely driven by the function (or loss of) *SMAD4*, but likely driven by additional genetic, epigenetic, and environmental factors.

The lack of in vivo correlation between *SMAD4* and the basal-like expression subtype (or *GATAT6* low expression) warrants further discussion. Preclinical data suggested that *SMAD4* can regulate the expression of *GATA6*. Using hTERT immortalized pancreatic ductal epithelial cells, suppressed *SMAD4* (via small interfering RNA) reduced *GATA6* expression. Conversely, FLAG-SMAD4 overexpression in PSN1 cells (which are SMAD4 deleted) resulted in re-establishment of *GATA6* [10]. However, these findings did not translate to the human PDAC specimens included in our study. The absence of a direct molecular correlation between these two biomarkers diminishes the likelihood of a single, druggable target to address treatment resistance. Moreover, the clinical significance of the basal-like subtype, regardless of *SMAD4* mutational status, further emphasizes the divergence of these molecular pathways. In these data, the basal-like subtype was associated with trends to inferior survival in the TCGA and significantly inferior survival in the CPTAC, and such associations were not impacted by the loss of *SMAD4*. While the association between basal-like subtype or low *GATA6* and inferior survival are well established in the literature [7,8,10,13], the impact (or lack thereof) of concurrent *SMAD4* alterations represents an additional complexity to this search for precision oncology.

*GATA6*, a member of the transcription factor family, binds to the (A/T)GATA(A/G) consensus sequence, influencing gene expression [34]. Crucial for cell differentiation, *GATA6* is vital for maintaining the exocrine pancreas [35]. Recent studies propose a tumor-suppressive role of *GATA6* in PDAC mouse models, influencing both differentiation and cancer-related transcriptional programs [36,37]. In human PDAC cells, *GATA6* plays a pivotal role in inhibiting de-differentiation and epithelial–mesenchymal transition (EMT). The sequential regulation of EMT and mesenchymal–epithelial transition (MET) is crucial for effective tumor spreading [38,39]. In human PDAC samples, the loss of *GATA6* in PDAC primary samples correlates with altered differentiation and shorter overall patient survival [40,41].

Our study is not without limitations. Conducting a retrospective analysis of clinical outcomes using a database inherently carries the risk of selection bias and potential inaccuracies in data reporting. Additionally, there is the possibility of sampling error in tumor biopsies, which could lead to mischaracterizations of *SMAD4* mutational status or *GATA6* mRNA expression. This is a particular limitation of genomic classification of PDAC given the large stromal component of many tumors. Third, certain analyses may have been underpowered given the limited sample sizes. The basal-like phenotype was associated with statistically inferior survival in the CPTAC cohort but not in the TCGA cohort; we believe this represents a type II error in the TCGA cohort. We methodologically avoided combining the TCGA and CPTAC datasets given the differences in tumor processing, which may have also contributing to some conflicting results. Fourth, the survival analyses may have been influenced by heterogeneity in the use of adjuvant therapy, for which data were not available. Additionally, analyses of recurrence-free survival were limited by the lack of such data in these cohorts.

Nonetheless, our study represents the largest cohort of PDAC patients with direct evaluation of the relationship between *SMAD4* mutation and the basal-like subtype/*GATA6* expression. Given the lack of correlation, distinct biomarker-driven clinical trials and individualized studies exploring the mechanistic basis of each biomarker are necessary for the advancement of precision oncology for this disease.

## Figures and Tables

**Figure 1 biomedicines-11-03058-f001:**
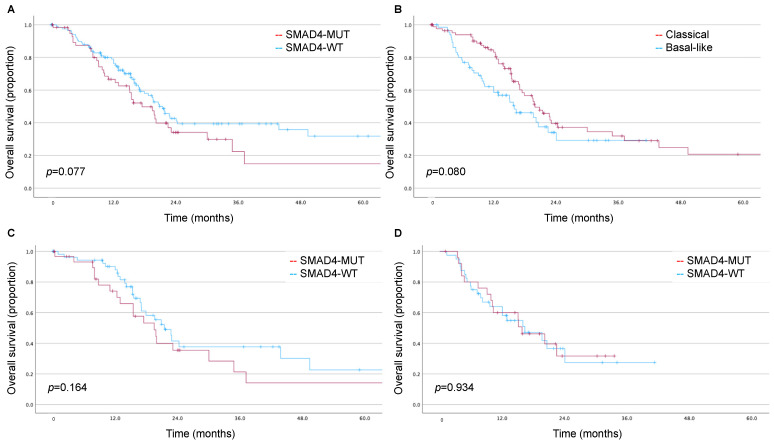
Kaplan–Meier estimates for the impact of *SMAD4* and molecular subtype on overall survival in the TCGA cohort. (**A**) Association of *SMAD4* alterations with overall survival; (**B**) association of Moffitt subtype with overall survival; (**C**) association of *SMAD4* alterations with overall survival in the TCGA subset defined by the classical subtype; (**D**) association of *SMAD4* alterations with overall survival in the TCGA subset defined by the basal-like subtype.

**Figure 2 biomedicines-11-03058-f002:**
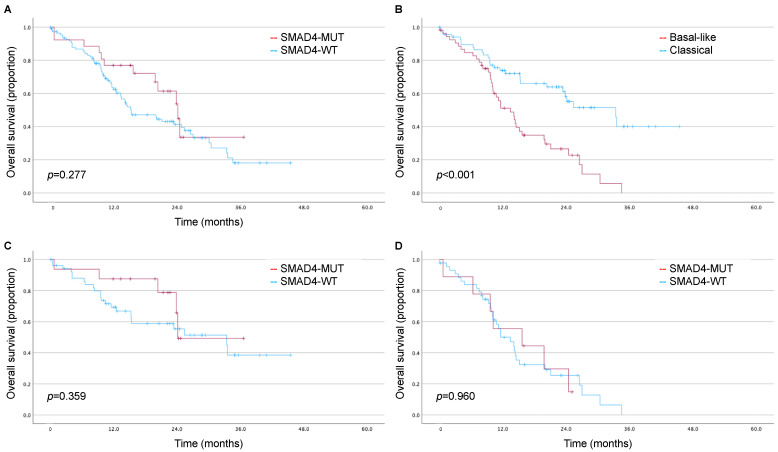
Kaplan–Meier estimates for the impact of *SMAD4* and molecular subtype on overall survival in the CPTAC cohort. (**A**) Association of *SMAD4* alterations with overall survival; (**B**) association of Moffitt subtype with overall survival; (**C**) association of *SMAD4* alterations with overall survival in the CPTAC subset defined by the classical subtype; (**D**) association of *SMAD4* alterations with overall survival in the CPTAC subset defined by the basal-like subtype.

**Table 1 biomedicines-11-03058-t001:** Clinicodemographics of TCGA Patient Cohort.

	SMAD4-WT		SMAD4-MUT		*p*
	(n = 120)		(n = 60)		
	#	%	#	%	
Age					0.874
Median (IQR)	65 (54–74)		65 (58–73)		
Sex					0.751
Female	55	45.8%	26	43.3%	
Male	65	54.2%	34	56.7%	
Race ^a^					0.041
White	106	90.6%	52	88.1%	
Black	6	5.1%	0	0.0%	
Asian	5	4.3%	7	11.9%	
Ethnicity ^b^					0.588
Hispanic	3	2.5%	2	3.3%	
AJCC Stage ^c^					0.412
I	16	13.6%	6	10.0%	
II	95	80.5%	53	88.3%	
III	4	3.4%	0	0.0%	
IV	3	2.5%	1	1.7%	
T Stage					0.596
T1	5	4.2%	3	5.0%	
T2	16	13.3%	7	11.7%	
T3	94	78.3%	50	83.3%	
T4	3	2.5%	0	0.0%	
TX	2	1.7%	0	0.0%	
N Stage					0.839
N-negative	35	29.2%	15	25.0%	
N-positive	81	67.5%	43	71.7%	
NX	4	3.3%	2	3.3%	
Grade					0.373
G1	24	20.0%	7	11.7%	
G2	61	50.8%	35	58.3%	
G3	33	27.5%	18	30.0%	
GX	2	1.7%	0	0.0%	

Abbreviations: IQR interquartile range; AJCC American Joint Committee on Cancer. ^a^ Race data unavailable for 4 patients. ^b^ Ethnicity data unavailable for 43 patients. ^c^ AJCC stage data unavailable for 2 patients.

**Table 2 biomedicines-11-03058-t002:** Clinicodemographics of CPTAC Patient Cohort.

	SMAD4 Wild-Type		SMAD4 Mutant		*p*
	(n = 115)		(n = 26)		
	#	%	#	%	
Age					0.147
Median (IQR)	64 (59–71)		66 (63–72)		
Sex					0.556
Female	56	48.7%	11	42.3%	
Male	59	51.3%	15	57.7%	
AJCC Stage ^a^					0.491
I	18	16.4%	5	20.8%	
II	48	43.6%	12	50.0%	
III	35	31.8%	7	29.2%	
IV	9	8.2%	0	0.0%	
T Stage					0.799
pT1	7	6.1%	3	11.5%	
pT2	72	62.6%	17.2	57.7%	
pT3	33	28.7%	19.5	30.8%	
pT4	1	0.9%	0	0.0%	
pTX	2	1.7%	0	0.0%	
N Stage					0.842
pN0	24	20.9%	7	26.9%	
pN1	44	38.2%	10	38.5%	
pN2	40	34.8%	7	26.9%	
pNX	7	6.1%	2	7.7%	
LVI ^b^					0.429
Absent	31	29.2%	9	37.5%	
Present	75	70.8%	15	62.5%	
PNI ^c^					0.471
Absent	15	13.8%	2	8.3%	
Present	94	86.2%	22	91.7%	

Abbreviations: IQR interquartile range; AJCC American Joint Committee on Cancer; LVI lymphovascular invasion; PNI perineural invasion. ^a^ AJCC stage data unavailable for 7 patients. ^b^ LVI data unavailable for 11 patients. ^c^ PNI data unavailable for 8 patients.

**Table 3 biomedicines-11-03058-t003:** Association between SMAD4 alterations and GATA6 expression and Moffitt subtype.

	SMAD4 Wild-Type		SMAD4 Mutant		*p*
	#	%	#	%	
*TCGA*					
GATA6 Tertile					0.928
Low	41	34.2%	19	31.7%	
Medium	40	33.3%	20	33.3%	
High	39	32.5%	21	35.0%	
Moffitt Subtype ^a^					0.783
Classical	54	57.4%	32	55.2%	
Basal-like	40	42.6%	26	44.8%	
*CPTAC*					
GATA6 Tertile					0.828
Low	39	31.5%	8	30.8%	
Medium	39	31.5%	8	30.8%	
High	46	37.0%	10	38.4%	
Moffitt Subtype ^b^					0.416
Classical	55	55.0%	16	64.0%	
Basal-like	45	45.0%	9	36.0%	

^a^ Moffitt subtype unavailable for 28 patients in TCGA cohort. ^b^ Moffitt subtype unavailable for 16 patients in CPTAC cohort.

## Data Availability

Data are publicly available on cbioportal.

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
