# Peer review of "Association between SMAD4 Mutations and GATA6 Expression in Paired Pancreatic Ductal Adenocarcinoma Tumor Specimens: Data from Two Independent Molecularly-Characterized Cohorts"

_biomedicines, 2023, doi:10.3390/biomedicines11113058_

Round 1

Reviewer 1 Report

Comments and Suggestions for Authors

The study aimed to explore the relationship between two molecular biomarkers, SMAD4 mutations and GATA6 expression, in localized pancreatic ductal adenocarcinoma (PDAC) tumors. To achieve this, molecular data from 321 patients across two datasets, TCGA (n=180) and CPTAC (n=141), were analyzed. The study concludes that since there was no observed association between SMAD4 mutations and GATA6 expression or between SMAD4 mutations and Moffitt subtype, further targeted research might be needed to understand the distinct mechanisms and implications of each biomarker in PDAC.

In the abstract, the authors should clearly describe the objective and provide background on the role of these two genes in pancreatic cancer and their influence on treatment outcomes. Was there a significant correlation found between SMAD4 mutations and GATA6 expression in these two cohorts? A mention of the established relationship between SMAD4 and GATA6 in pancreatic cancer treatments would elucidate the existing knowledge gap. While the study concludes that no observed association exists between the biomarkers, the authors should delve into the potential therapeutic or clinical implications based on their findings.

The potential molecular mechanisms or pathways responsible for the observed findings should be further elucidated, as such insights have vital therapeutic implications. The apparent lack of correlation between SMAD4 and the basal-like subtype in PDAC, although of potential significance, requires further validation through more stringent statistical analyses, such as Cox proportional-hazards regression models, multivariate regression analysis, etc.

Author Response

Manuscript ID: biomedicines-2676432

Association Between SMAD4 Mutations and GATA6 Expression in Paired Pancreatic Ductal Adenocarcinoma Tumor Specimens

Dear Dr. Mousa,

Thank you for the opportunity to submit our revised manuscript entitled “Association Between SMAD4 Mutations and GATA6 Expression in Paired Pancreatic Ductal Adenocarcinoma Tumor Specimens” to Biomedicines. We thank the reviewer for their critical appraisal of our manuscript. We now present a revised manuscript, incorporating the reviewer’s suggestions. All changes have been highlighted in this revision. A detailed response to reviewer comments follows. We hope that our revisions will resolve their concerns and questions.

Sincerely,

Brett L. Ecker, MD

Rutgers Cancer Institute of New Jersey

Rutgers Robert Wood Johnson Medical School

Cooperman Barnabas Medical Center

94 Old Short Hills Road | Suite 1172 | Livingston | NJ 07039

(P) 973.322.5195 | (F) 973-322-2471

Reviewer #1:

1. The study aimed to explore the relationship between two molecular biomarkers, SMAD4 mutations and GATA6 expression, in localized pancreatic ductal adenocarcinoma (PDAC) tumors. To achieve this, molecular data from 321 patients across two datasets, TCGA (n=180) and CPTAC (n=141), were analyzed. The study concludes that since there was no observed association between SMAD4 mutations and GATA6 expression or between SMAD4 mutations and Moffitt subtype, further targeted research might be needed to understand the distinct mechanisms and implications of each biomarker in PDAC.

Response: Thank you for your thorough review of our manuscript.

2. In the abstract, the authors should clearly describe the objective and provide background on the role of these two genes in pancreatic cancer and their influence on treatment outcomes. Was there a significant correlation found between SMAD4 mutations and GATA6 expression in these two cohorts? A mention of the established relationship between SMAD4 and GATA6 in pancreatic cancer treatments would elucidate the existing knowledge gap. While the study concludes that no observed association exists between the biomarkers, the authors should delve into the potential therapeutic or clinical implications based on their findings.

Response: Thank you for your feedback. We agree and have added to the abstract to provide further background information, namely that each has been associated with inferior survival uniquely for patients receiving 5-FU-based therapies, and that SMAD4 may directly regulate the expression of GATA6 in PDAC, pointing to a common predictive biomarker. Contrary to hypothesis, there was a lack of association between these two biomarkers in either cohort in our study. We have updated the abstract to reflect the clinical implications of these data, namely that distinct biomarker-defined clinical trials are necessary.

3. The potential molecular mechanisms or pathways responsible for the observed findings should be further elucidated, as such insights have vital therapeutic implications. The apparent lack of correlation between SMAD4 and the basal-like subtype in PDAC, although of potential significance, requires further validation through more stringent statistical analyses, such as Cox proportional-hazards regression models, multivariate regression analysis, etc.

Response: We agree and have added a multivariable Cox regression model adjusting for known prognostic variables that were significantly associated with survival in each cohort (i.e., AJCC stage, N stage and Grade for the TCGA cohort; and AJCC stage for the CPTAC Cohort). SMAD4 was not independently associated with survival in these models.  The adjusted hazard ratios have been added to the manuscript.

Reviewer 2 Report

Comments and Suggestions for Authors

Joshua D. Greendyk BA et al showed that SMAD4 mutations and GATA6 expression in paired pancreatic ductal adenocarcinoma tumor specimens.

The manuscript is interesting. However, there are many shortcomings to explain the whole of the paper.

It would have been easier to understand if it was written in a graph rather than in a table (Table 1 and Table 2). For example, if authors show a graph that can be compared by age group or race type, I think it will be effectively compared. 

Based on the information of a large number of patients, I think this is a useful paper that shows the relationship between SMAD4 and GATA6 in the pancreatic ductal adenocarcinoma tumor.

Comments on the Quality of English Language

Minor editing of English language required

Author Response

Reviewer #2:

4. Joshua D. Greendyk BA et al showed that SMAD4 mutations and GATA6 expression in paired pancreatic ductal adenocarcinoma tumor specimens. The manuscript is interesting. Based on the information of a large number of patients, I think this is a useful paper that shows the relationship between SMAD4 and GATA6 in the pancreatic ductal adenocarcinoma tumor. However, there are many shortcomings to explain the whole of the paper. It would have been easier to understand if it was written in a graph rather than in a table (Table 1 and Table 2). For example, if authors show a graph that can be compared by age group or race type, I think it will be effectively compared. Based on the information of a large number of patients, I think this is a useful paper that shows the relationship between SMAD4 and GATA6 in the pancreatic ductal adenocarcinoma tumor.

Response: Thank you for your feedback. We have provided the clinicodemographic data, stratified by the presence versus absence of SMAD4 alterations, in the standard fashion of a “Table 1”.  We have provided figures for the display of survival curves.  We hope you agree that this is the standard scientific reporting of clinical data.

In distinction, the data of Table 3, which represents the primary analysis of this manuscript, could be presented as either a figure (e.g., bar graph) or table.  We would be happy to update the display of this analysis at your request.

Reviewer 3 Report

Comments and Suggestions for Authors

The rationale for studying SMAD4 variants versus GATA6 expression is not well explained. Is there any molecular link on for example transcriptional level? Thu study subject does not make a coherent story.

Some % did not add up to 100% in the Tables.

The authors should present some of their own data e.g. tissue microarray versusus variants. This is just a relatively low-dimensional analysis of publically available data with mostly negative results. This is too little work to make proper publication.

The study must be enhanced with a cohort based on another approach.

This is more like preliminary data or a poster.

Is GATA6 mutated in pancreatic cancer?

What about multivariate survival analysis with all relevant data?

Author Response

5. The rationale for studying SMAD4 variants versus GATA6 expression is not well explained. Is there any molecular link on for example transcriptional level? The study subject does not make a coherent story.

Response: Thank you for your feedback.  In agreement, we have updated the introduction to reflect that SMAD4 alterations, which are usually loss-of-function mutations, can directly influence the expression of GATA6. Both SMAD4 alterations and GATA6 expression are implicated in responsiveness to 5FU-based therapies.

6. Some % did not add up to 100% in the Tables.

Response: In agreement, we have updated the tables to have percentages calculated according to columns, rather than rows.

7. The authors should present some of their own data e.g. tissue microarray versus variants. This is just a relatively low-dimensional analysis of publicly available data with mostly negative results. This is too little work to make proper publication. The study must be enhanced with a cohort based on another approach. This is more like preliminary data or a poster.

Response: This is a good point. We looked to include our institutional mutational data but the Rutgers University cohort size was smaller than either of these publicly available cohorts.  Moreover, the clear lack of association between SMAD4 alterations and GATA6 expression in these two cohorts is unlikely to benefit from increased statistical power.  The study aim was to evaluate whether SMAD4 and GATA6 are related versus unique biomarkers, which has immediate clinical implications in randomized trial design.  Ongoing clinical trials (e.g., COMPASS trial; NCT02750657) are exploring GATA6 as a predictive biomarker, yet it was unknown if additional trials using SMAD4 as a biomarker would be redundant.  These data point to the need for independent clinical trials incorporating each biomarker.  The manuscript has been updated to reflect the immediate clinical implications of these data on trial design.

8. Is GATA6 mutated in pancreatic cancer?

Response: GATA6 mutations are rare in PDAC tumors. We observed 0 (0%) GATA6 mutations in TCGA and 1(0.7%) mutations in the CPTAC cohort.  This rarity limited further analyses.

9. What about multivariate survival analysis with all relevant data?

Response: We agree and have added a multivariable Cox regression model adjusting for known prognostic variables that were significantly associated with survival in each cohort (i.e., AJCC stage, N stage and Grade for the TCGA cohort; and AJCC stage for the CPTAC Cohort). SMAD4 was not independently associated with survival in these models.   The adjusted hazard ratios have been added to the manuscript.

Reviewer 4 Report

Comments and Suggestions for Authors

With pleasure, I read the paper titled “Association Between SMAD4 Mutations and GATA6 Expression in Paired Pancreatic Ductal Adenocarcinoma Tumor Specimens”. The topic is clinically relevant to practice, and of importance to the readers of Biomedicine journal. Overall, the manuscript reads well and has good flow of ideas, up-to-date citations, and proper summary of data using tables and figures. A MAJOR strength of the article is being—according to authors, the most comprehensive exploration of SMAD4 and GATA6 in PDAC. The research is well-articulated to encourage more research in the field. The introduction section was detailed enough to provide the reader with the needful background information. The methods section was detailed too, however, some edits are needed for complete reporting. The discussion section comprised elaborations from clinical and biological aspects, adding intellectual curiosity. The research had some unavoidable limitations, all of which had been explicitly acknowledged. The conclusion is line with the presented results. All in all, this manuscript is clinically significant and is very likely to be cited extensively in the future. I strongly recommend the manuscript to be accepted—however, prior to that, some combinatorial minor and major changes are required as indicated below:

TITLE

— I recommend changing title to: “Association Between SMAD4 Mutations and GATA6 Expression in Paired Pancreatic Ductal Adenocarcinoma Tumor (PDAC) Specimens: Data from two independent molecularly-characterized PDAC cohorts”

ABSTRACT

— It is important to mention that SMAD4-mutation did not appear to associate with poor clinico-pathological features, such as poor tumor grade, advanced tumor stage, positive LVSI, and positive PNI compared with SMAD4-wildtype.

INTRODUCTION

— Please clarify the nature of SMAD4 alterations associated with unfavorable clinical outcomes. Are they loss-of-function or gain-of-function mutations?

— You mentioned that “SMAD4 may directly regulate the expression of GATA6”. Is it true for the opposite; does GATA6 directly regulate SMAD4?

— Please clearly highlight the significance of your research. Is this the first-ever study to investigate the relationship between SMAD4 and GATA6 in PDAC? If not, please mention how does your research enriches existing literature.

— Please conclude the section with some proposed hypotheses.

METHODS

— Please mention how the tertiles for GATA6 into low, medium, and high were established?

— It would be interesting to investigate also the rates of GATA6 mutations (WT versus MUT), if any, and investigate its impact on clinicopathological features and overall survival.

— You investigated the GATA6 expression at the mRNA level. I wonder if the proteomic data are available and whether the mRNA and protein levels are matched in the samples.

— You explored the outcome of overall survival, and I wonder if it is possible to enrich the manuscript by investigating also the disease-free survival or recurrence-free survival?

RESULTS

— In Table 1 and Table 2, the percentages were calculated according to rows. However, this is not right. The percentages should be calculated according to columns. Considering the data in Table 1 and Table 2, does it mean that SMAD4-mutation does not appear to associate with poor clinico-pathological features, as there was no difference in tumor grade, tumor stage, LVSI, and PNI between SMAD4-WT and SMAD4-MUT?

— In Table 3, the percentages were calculated according to rows. However, this is not right. The percentages should be calculated according to columns. The data from TCGA and CPTAC showed no impact of SMAD4-mutation on GATA6 expression and Moffitt subtype—were these observations predicted, and what are your thoughts?

— The TCGA data showed no difference in OS based on histological subtypes. However, the CPTAC data showed worse overall survival for basal-like type compared with the classical type. Why was there a discrepancy in findings between the two datasets?

— It is possible to mention the type of chemotherapeutic regimens received by the patients in TCGA and CPTAC datasets? 

DISCUSSION

— Please provide some biological discussion on the role of GATA6 and how its low expression negatively influences the phenotype and prognosis of PDAC.

— Are there any in-vitro studies that examined the relationship between SMAD4 and GATA6 in PDAC? If so, please briefly summarize their findings to enrich the discussion section from a biological perspective.

OVERALL

— The manuscript requires editing for English language.

Comments on the Quality of English Language

Minor English editing is needed

Author Response

10. With pleasure, I read the paper titled “Association Between SMAD4 Mutations and GATA6 Expression in Paired Pancreatic Ductal Adenocarcinoma Tumor Specimens”. The topic is clinically relevant to practice, and of importance to the readers of Biomedicine journal. Overall, the manuscript reads well and has good flow of ideas, up-to-date citations, and proper summary of data using tables and figures. A MAJOR strength of the article is being—according to authors, the most comprehensive exploration of SMAD4 and GATA6 in PDAC. The research is well-articulated to encourage more research in the field. The introduction section was detailed enough to provide the reader with the needful background information. The methods section was detailed too, however, some edits are needed for complete reporting. The discussion section comprised elaborations from clinical and biological aspects, adding intellectual curiosity. The research had some unavoidable limitations, all of which had been explicitly acknowledged. The conclusion is line with the presented results. All in all, this manuscript is clinically significant and is very likely to be cited extensively in the future. I strongly recommend the manuscript to be accepted—however, prior to that, some combinatorial minor and major changes are required as indicated below:

Response: Thank you for your thorough review of our manuscript.

11. TITLE — I recommend changing title to: “Association Between SMAD4 Mutations and GATA6 Expression in Paired Pancreatic Ductal Adenocarcinoma Tumor (PDAC) Specimens: Data from two independent molecularly-characterized PDAC cohorts”

Response: In agreement, the manuscript title has been updated.

12. ABSTRACT — It is important to mention that SMAD4-mutation did not appear to associate with poor clinico-pathological features, such as poor tumor grade, advanced tumor stage, positive LVSI, and positive PNI compared with SMAD4-wildtype.

Response: In agreement, we have updated the abstract to include the clinicopathologic features stratified by SMAD4 mutational status.

13. INTRODUCTION — Please clarify the nature of SMAD4 alterations associated with unfavorable clinical outcomes. Are they loss-of-function or gain-of-function mutations?

Response: Thank you for pointing this out, the manuscript has been updated to clarify that typically loss-of-function mutations are observed.

14. — You mentioned that “SMAD4 may directly regulate the expression of GATA6”. Is it true for the opposite; does GATA6 directly regulate SMAD4?

Response: This is an interesting question.  For our analyses, we focused on SMAD4 alterations and not SMAD4 expression, as only the former has been identified as a prognostic biomarker.  Moreover, GATA6 mutations are rare in PDAC tumors. We observed 0 (0%) GATA6 mutations in TCGA and 1(0.7%) mutations in the CPTAC cohort. Thus, genomic alterations in GATA6 are unlikely drivers of SMAD4 expression.

15. — Please clearly highlight the significance of your research. Is this the first-ever study to investigate the relationship between SMAD4 and GATA6 in PDAC? If not, please mention how does your research enriches existing literature.

Response: Thank you, we have updated the introduction stating that this is the largest study to examine the relationship between SMAD4 and GATA6 in human PDAC samples.

16. — Please conclude the section with some proposed hypotheses.

Response: The introduction has been updated with the following: We hypothesized that SMAD4 alterations would be positively correlated with GATA6 expression and basal subtype.

17. METHODS — Please mention how the tertiles for GATA6 into low, medium, and high were established?

Response: The tertiles were established by equal grouping (within each study) based on normalized expression, reflecting the methodology used in previous work. The manuscript has been updated.

18. — It would be interesting to investigate also the rates of GATA6 mutations (WT versus MUT), if any, and investigate its impact on clinicopathological features and overall survival.

Response: GATA6 mutations are rare in PDAC tumors. We observed 0 (0%) GATA6 mutations in TCGA and 1(0.7%) mutations in the CPTAC cohort.  This rarity limited further analyses.

19. — You investigated the GATA6 expression at the mRNA level. I wonder if the proteomic data are available and whether the mRNA and protein levels are matched in the samples.

Response: This is an interesting exploratory aim that we would like to explore in a future manuscript.  We focused our analysis on the interaction of two known prognostic biomarkers (i.e., GATA6 mRNA expression/basal-like phenotype and SMAD4 alterations), with the immediate impact of such data on determining whether ongoing clinical trial design necessitates distinct biomarker-driven studies.  We agree that this is an interesting question, however the proteomic data is not available for this entire cohort. The TCGA cohort has missing protein expression (RPPA) for approximately one-third of patients.

20. — You explored the outcome of overall survival, and I wonder if it is possible to enrich the manuscript by investigating also the disease-free survival or recurrence-free survival?

Response: This is a great suggestion that we explored in the dataset.  Unfortunately, disease-free/recurrence-free survival data is missing for the majority of the TCGA cohort (110/180; 61.1%) and unavailable for all (100%) of the CPTAC cohort.  Data missingness prohibited further study.  We have updated our limitations section to reflect this point.

21. RESULTS — In Table 1 and Table 2, the percentages were calculated according to rows. However, this is not right. The percentages should be calculated according to columns. Considering the data in Table 1 and Table 2, does it mean that SMAD4-mutation does not appear to associate with poor clinico-pathological features, as there was no difference in tumor grade, tumor stage, LVSI, and PNI between SMAD4-WT and SMAD4-MUT?

Response: Thank you for pointing this out. Tables 1 and 2 have been updated to show the percentages calculated according to columns. The data does support the conclusion that SMAD4-mutation is not associated with poor clinico-pathological features, as there was no difference in tumor grade, tumor stage, LVSI, and PNI when compared to SMAD4-WT.

22. — In Table 3, the percentages were calculated according to rows. However, this is not right. The percentages should be calculated according to columns. The data from TCGA and CPTAC showed no impact of SMAD4-mutation on GATA6 expression and Moffitt subtype—were these observations predicted, and what are your thoughts?

Response: Thank you for pointing this out. Table 3 has been updated to show the percentages calculated according to columns. We had hypothesized that SMAD4 alterations would be positively correlated with low GATA6 expression and the basal-like phenotype.  Given the lack of signal in these large cohorts, these data support that these biomarkers are not functionally related in human PDAC specimens.  The Discussion has been updated to include this interpretation.

24. — The TCGA data showed no difference in OS based on histological subtypes. However, the CPTAC data showed worse overall survival for basal-like type compared with the classical type. Why was there a discrepancy in findings between the two datasets?

Response: The basal-like phenotype was associated with inferior survival in the CPTAC cohort (p<0.001).  In the TCGA cohort, the survival of patients with a basal-like phenotype tumor was inferior, although this did not reach statistical significance (p=0.080).  We believe this represents a type II error in the TCGA cohort. Alternatively, this biomarker may have variable prognostic utility.  We believe this is less likely given additional published studies that have observed inferior survival associated with the basal-like phenotype.  The limitations section of the Discussion has been updated accordingly.

25. — It is possible to mention the type of chemotherapeutic regimens received by the patients in TCGA and CPTAC datasets? 

Response: This is an excellent suggestion.  Unfortunately, these data are not available in these cohorts. We note in the limitations that the survival analyses may have been influenced by heterogeneity in the use of adjuvant therapy, for which data was not available. 

26. DISCUSSION — Please provide some biological discussion on the role of GATA6 and how its low expression negatively influences the phenotype and prognosis of PDAC.

Response: Thank you for your feedback. We have added two paragraphs to the discussion covering the biology of GATA6 and the influences on phenotype and prognosis of PDAC.

27. — Are there any in-vitro studies that examined the relationship between SMAD4 and GATA6 in PDAC? If so, please briefly summarize their findings to enrich the discussion section from a biological perspective.

Response: In agreement, we have expanded upon the in-vitro studies where SMAD4 directly regulate the expression of GATA6.  Using hTERT immortalized pancreatic ductal epithelial cells, suppressed SMAD4 (via small interfering RNA) reduced GATA6 expression.  Conversely, FLAG-SMAD4 overexpression in PSN1 cells (which are SMAD4 deleted) resulted in re-establishment of GATA6.

Round 2

Reviewer 1 Report

Comments and Suggestions for Authors

I have no further questions about this version. 

Reviewer 3 Report

Comments and Suggestions for Authors

I have the same general reservations as in the first review i.e. the study is generally little work. A few more studies like that could be generated by 1-2 people over a short period.

Still, the authors generally added the formally requested point to the manuscript and there are no major flaws.

Comments on the Quality of English Language

English is fine.

Reviewer 4 Report

Comments and Suggestions for Authors

The authors did a wonderful job by addressed all the comments adequately. The manuscript now reads well and is scientifically valid, methodologically robust, and intellectually interesting. The limitations have been properly acknowledged. All in all, I congratulate the authors on a well-done revision and the manuscript can be accepted in its current stage. 

Comments on the Quality of English Language

Minor editing my be required